# Optimizing Visual Question Answering Models for Driving: Bridging the Gap Between Human and Machine Attention Patterns

## Abstract

*Visual Question Answering (VQA) models play a critical role in enhancing the perception capabilities of autonomous driving systems by allowing vehicles to analyze visual inputs alongside textual queries, fostering natural interaction and trust between the vehicle and its occupants or other road users. This study investigates the attention patterns of humans compared to a VQA model when answering driving-related questions, revealing disparities in the objects observed. We propose an approach integrating filters to optimize the model's attention mechanisms, prioritizing relevant objects and improving accuracy. Utilizing the LXMERT model for a case study, we compare attention patterns of the pre-trained and Filter Integrated models, alongside human answers using images from the NuImages dataset, gaining insights into feature prioritization. We evaluated the models using a Subjective scoring framework which shows that the integration of the feature encoder filter has enhanced the performance of the VQA model by refining its attention mechanisms.*

## 1. Introduction

Visual Question Answering (VQA) models are integral to autonomous driving systems as they enable vehicles to perceive and understand their surroundings by analyzing visual inputs alongside textual queries, thereby enhancing their perception capabilities. VQA models facilitate natural interaction between the vehicle and its occupants or other road users, fostering trust in autonomous technology. By enabling natural language interaction, VQA models assist in making the autonomous vehicle more transparent and understandable to the driver. When the vehicle can effectively communicate its actions, intentions, and reasoning in a language that humans understand, it fosters a sense of transparency and predictability, which are crucial for building trust.

For instance, if the vehicle encounters a challenging driv-

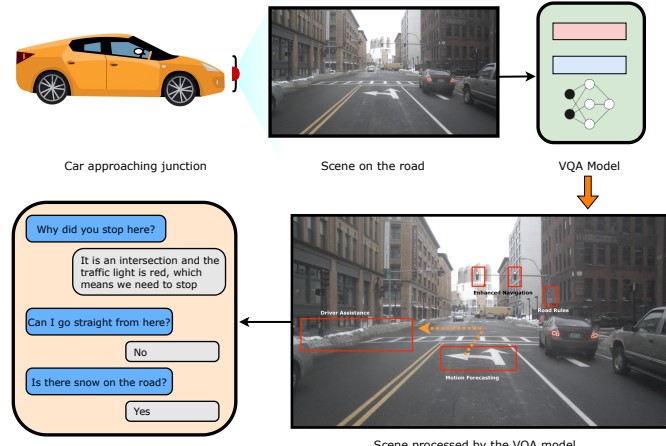

Figure 1. A demo of how VQA models work in a driving scenario

ing scenario, it can better explain its decision-making process to the driver using the VQA model. This allows the driver to better comprehend the situation and feel more confident in the vehicle's capabilities. Moreover, in situations where the driver needs clarification or wants to ask questions about the vehicle's actions or the environment, the VQA model can provide immediate responses, helping to alleviate uncertainties and concerns.

This study is focused on comparing the explanation given for object detection patterns of humans and attention patterns of a Visual Question Answering (VQA) model when answering questions related to driving. Our survey indicated that humans concentrate on objects like road lines, signboards, vehicles in the ego lane, etc when it comes to answering questions related to driving. However, when we looked at the objects observed by a VQA model, it wasn't restricted to only objects related to driving. There were objects like trees, sky, tower, etc which were irrelevant to answer a question like, "How many vehicles are in the ego lane?". The approach here is to streamline the features and objects that the VQA model is taking into consideration by adding a filter when asking a driving-related question. This will optimize the model's attention mechanisms to priori-

tize relevant objects and improve its accuracy in answering questions.

It also addresses a disparity between human attention patterns and those of the VQA model, aiming to enhance the model's performance in the domain of driving. We are performing a case study with a VQA model- LXMERT where we look at how the pretrained model with all its features answers a driving question and how a 'filter' integrated model answers the same question while also comparing them with the Human Answers that were provided by human annotators. By comparing the attention patterns of the pretrained and streamlined model, we can gain insights into how different features and objects are prioritized when answering driving-related questions. This analysis can help in identifying the factors that contribute to the models' performance differences.

By examining attention mechanisms, we aim to elucidate how VQA models prioritize visual stimuli in their decision-making processes which will help us in the finetuning process of our experiments.

## 2. Background Study

Vision transformers in a Visual Question Answering (VQA) model work by dividing the image into patches and representing them as embeddings [5]. These embeddings, along with the text embeddings of the question, are then fed into a transformer architecture [5]. The transformer processes the embeddings by attending to both visual and textual information, enabling the model to understand the image and the question simultaneously. Finally, the model generates an answer based on the learned representations from the transformer layers. In [5], the authors argue that uncertainty in vision is a dominating factor preventing the successful learning of reasoning in vision and language problems. By integrating a filter that focuses on driving-related features, our approach aims to mitigate this uncertainty by providing a VQA model with more relevant visual information tailored to the context of driving-related questions.

In [12], they focus on improving the efficiency of visual transformers by removing redundant calculations in transformer networks. Considering that the attention mechanism in a transformer architecture aggregates different patches layer-by-layer, the authors Yehui Tang et al. present a novel 'patch slimming' approach that discards useless patches in a top-down paradigm. Initially, the effective patches in the last layer are identified and then used to guide the patch selection process of previous layers. For each layer, the impact of a patch on the final output feature is approximated and patches with less impact will be removed [12]. While this could work for a vision transformer model, it is not necessarily good to implement for a VQA model. Patch slimming aims to improve the efficiency of the model by removing redundant patches throughout the image and the

filter focuses on extracting driving-related features from the image before passing it through the vision transformer, to enhance the model's ability to answer driving-related questions more effectively [12]. The impact of patch slimming on performance can be more general, affecting the overall efficiency of the model but potentially risking loss of task-specific information [12]. However, integrating a filter focusing on driving-related features directly aims to enhance performance on driving-related questions by ensuring that the model receives relevant visual information. Patch slimming is a more general approach that may not adapt specifically to the requirements of the VQA task, which can result in a loss of task-specific information. Integrating a filter specifically designed for driving-related questions ensures that the model prioritizes relevant features for this task, leading to improved performance on driving-related questions while maintaining task specificity.

In [7], a novel object detection framework is proposed that attempts to extract meaningful and representative features across different image scales. The authors do so by unifying atrous convolutions with a vision transformer (DIL-ViT). The proposed model uses atrous convolutions to generate rich multi-scale feature maps and employs a self-attention mechanism to enrich important backbone features [7]. This framework enhances object detection performance which could be an excellent feature to add to a VQA model. However, in our case, the VQA model in question has to enhance its performance on driving-related questions by ensuring that the model receives relevant visual information. The filter proposed in our work specifically extracts driving-related features from the input image before passing it through the vision transformer component of the VQA model. While both the framework proposed in [7] and the filter integration approach aim to enhance model performance, they differ in their focus, purpose, task applicability, feature extraction methods, training objectives, and adaptation requirements.

In [14], the authors argue that the existing methods suffer from bias in understanding the image and insufficient knowledge to solve the problem of VQA. The authors propose a novel knowledge-based VQA framework (PROOFREAD) that uses LLM to obtain knowledge explicitly and the vision language model which can see the image to get the knowledge answer and a knowledge perceiver that filters out knowledge that is deemed harmful for getting the correct final answer [14]. PROOFREAD processes textual knowledge obtained by a language model, filtering out irrelevant or harmful information before combining it with the visual information [14] whereas our filter focuses on processing visual information from the image, extracting driving-related features, and integrating them into the VQA model's processing pipeline before combining them with textual information. The framework in [14] is designed to

address biases in understanding images and insufficiencies in knowledge to solve VQA problems in general whereas the filter proposed is tailored for improving VQA performance on driving-related questions specifically, focusing on extracting features relevant to driving scenarios from the image input.

## 3. Proposed Methodology

In Visual Question Answering (VQA) models, the model initially encodes the textual question into a numerical representation to capture its semantic meaning. As the inquiry pertains to a corresponding image, the model extracts visual features using convolutional neural networks (CNNs). Subsequently, the feature extraction mechanism dynamically assigns weights to different regions of the image or words in the question based on their relevance to the inquiry. This weighting enables the model to selectively focus on informative elements while disregarding irrelevant ones. Integrating the weighted features from both the image and question encoding, typically through concatenation or element-wise multiplication, the model combines visual and textual information. Finally, the integrated features are fed into a classifier to predict the answer, leveraging the learned associations between input features and corresponding answers from training data. Through this process, the attention pattern of the VQA model adapts to the specific question context, facilitating accurate and contextually relevant responses across a diverse range of topics.

While the architecture of a VQA model aims to replicate human cognition and reasoning when responding to inquiries about various scenarios, there exists a gap that requires attention. Typically, during driving, humans exhibit focused attention on aspects directly related to driving, often disregarding peripheral details unrelated to the task at hand [13]. When behind the wheel, individuals prioritize observing their immediate surroundings and assessing the next steps in their driving manoeuvres. This selective attention ensures optimal performance and safety on the road. For instance, if asked a question 'Is there snow on the road?' while driving, the driver's attention would primarily be directed towards assessing road conditions. They would observe the road surface for any signs of snow, focusing solely on elements pertinent to their driving task. This focused attention highlights a fundamental distinction between human perception during driving and the holistic scene understanding performed by VQA models. Therefore, bridging this gap necessitates the creation of a filter that enables the model to prioritize relevant information similar to human attentional patterns, thereby enhancing its ability to discern and respond accurately to questions posed in diverse real-world driving contexts.

### 3.1. Object Perception and Cognitive Processes

When presented with a question about a driving scenario, humans instinctively assess various factors to formulate a response. They consider the context, including details like location, weather, and traffic conditions, while also identifying potential hazards such as other vehicles, pedestrians, or adverse road conditions [13]. Drawing on their knowledge of traffic rules and regulations, they analyze the scenario through the lens of right-of-way, speed limits, and relevant guidelines [9]. Decision-making involves weighing the available options against safety, efficiency, and legal considerations, with a keen spatial awareness guiding their understanding of distances and relative speeds. Throughout this process, safety remains the most important concern, leading to actions aimed at minimizing risks and promoting responsible driving behaviour [13].

This complicated process has to be kept in mind while designing an autonomous driving system. These learnings also need to be incorporated into a VQA model if we want it to answer all our questions related to driving. Achieving this requires a deep understanding of human attention patterns, which can then be mirrored in the attention mechanisms of VQA models. We discuss in the following sections how aligning these attention patterns can improve the effectiveness of VQA models in handling driving-related inquiries.

#### 3.1.1 Human Answer Explanation Patterns

To gain an insight into the factors humans consider when given a driving scenario and posed with a question, we surveyed ten individuals with a minimum of five years of driving experience. Participants were asked to provide answers to questions depicted in Figures Tab. 1 and Sec. 5. The responses with the highest number of votes were selected as the definitive answers.

The features observed via answers to these questions were all cumulated together by asking the humans about the features observed using the same questionnaire. This explanation of features observed while making the decision to answer the given question helped us understand the recurring attention patterns in human observation and also compile a list of features that are commonly useful in answering driving-related questions.

#### 3.1.2 Attention Patterns of VQA models

The attention mechanism in a Visual Question Answering (VQA) model typically shows the focus or weight assigned to different regions of an image. Specifically, it indicates which parts of the input (such as image features or words in the question) are deemed most relevant or informative for answering the given question. By visualizing the attention

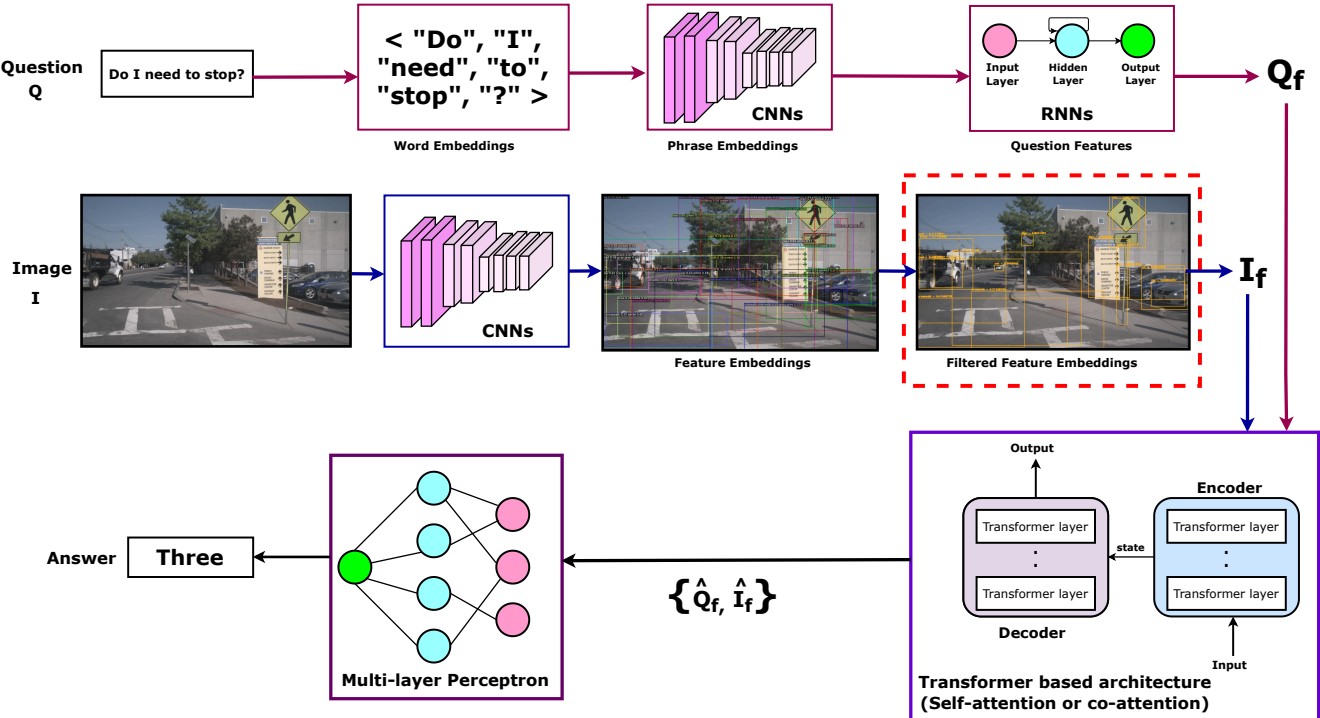

Figure 2. Refining VQA architecture: Integration of the filter into a general VQA architecture

weights, we can discern which areas of the image or words in the question the model prioritizes in its decision-making process. This helps in understanding the reasoning behind the model's responses and provides insights into how it processes and interprets visual and textual information to generate answers.

## 3.2. Human-Guided Feature Filter

We cumulated the features that are being observed by humans when answering driving-related questions and incorporated them in the construction of a filter aimed at capturing pertinent visual information. The objects like roads, lines, curbs, sidewalks, crosswalks, bikes, cars, trucks, etc., are recurring features in any given driving scenario which were used when creating the filter. This filter is designed to be integrated before the vision transformer component of the VQA model, ensuring that it focuses solely on relevant driving-related features as shown in Figure 2. This approach mimics human attention patterns, thereby enhancing the model's ability to effectively answer questions about driving scenarios by prioritizing the most relevant visual cues.

Filtering out irrelevant visual data reduces computational complexity and memory requirements, making the model more efficient and faster in processing information. This filter aligns the model's attention with human observation patterns, and the reasoning behind its predictions becomes more interpretable and aligned with human intuition. By emphasizing commonly observed features, it is observed in Case Studies (Section 4) that a VQA model can generalize better to new or unseen driving scenarios, enhancing its robustness and applicability in real-world settings. Prioritizing relevant visual cues related to driving can improve the safety and reliability of autonomous driving systems, ensuring they focus on critical information for making informed decisions on the road.

## 3.3. Filter: Algorithm and Need

The holistic approach typically employed by VQA models to capture and utilize intricate data patterns appears ineffective when narrowing the focus solely to driving-related questions. Thus, a filter is necessary to prevent the VQA model from expending computational resources on irrelevant learnings.

Integrating a filter before the vision transformer component of the VQA model helps to improve the model's performance when asked driving-related questions, as detailed further in Section 4. The advantages of this filter are listed as follows:

- **Feature Relevance:** By incorporating a filter specifically designed to capture driving-related features, the model can prioritize and emphasize information relevant to driving tasks. This can help the model to better focus on important visual cues such as road signs, vehicles, lanes, traffic lights, and road conditions, which are crucial for understanding and answering driving-related questions.

- **Reduced Noise:** Filtering out irrelevant visual information can help reduce noise in the input data, providing the model with cleaner and more focused inputs. This can prevent the model from being distracted by non-driving-related elements in the image, leading to more accurate predictions for driving-related questions.
- **Improved Attention Mechanism:** By pre-processing the input with a filter targeting driving-related features, the attention mechanism within the vision transformer component can be guided to attend more effectively to relevant regions of the image. This can enhance the model's ability to extract and leverage important visual information when generating answers to driving-related questions.
- **Enhanced Generalization:** Focusing the model's attention on driving-related features during pre-processing can help improve its generalization capabilities, allowing it to handle better variations in driving scenarios, lighting conditions, and camera perspectives. This can lead to more robust performance across different driving-related question types and real-world conditions.

#### 3.3.1 Algorithm

The filter proposed is shown in Algorithm **??**. It filters out irrelevant predictions based on predefined classes, extracts relevant information from the filtered predictions, converts the data to suitable types, and returns the filtered features.

---

**Algorithm 1:** Feature filter for Vision Transformer block in a VQA model

---

1 **Input:** Extract predicted classes, scores, bounding boxes, normalized bounding boxes, and ROI features from outputs tensor;
2 **Output:** Filtered features for VQA;
3 Initialize empty lists for filtered boxes, classes, labels, indices, normalized bounding boxes, and ROI features;
4 **if** *predicted class is in a predefined list of classes* **then**
5     Append the box, class, label, index, normalized bounding box, and ROI feature to the corresponding lists;
6 Convert filtered boxes, normalized bounding boxes, and ROI features to suitable data types;
7 **Return:** filtered boxes, classes, labels, indices, normalized bounding boxes, and ROI features;

---

This process helps in focusing the model's attention on the most relevant visual features for answering questions, thereby improving the overall performance of the VQA model.

## 4. Case Study

We perform a case study by incorporating the filter into a VQA model and observing the different answers before and after the filter is integrated. By comparing the model's responses before and after the filter's integration, we gain a clear understanding of the enhancements brought about by focusing on relevant driving-related features. We examine how the model's attention patterns evolve post-filter integration and can discern whether they align more closely with human observation patterns in driving scenarios. It is observed that this alignment enhances the model's ability to answer driving-related questions accurately along with their interpretability and generalization capabilities.

### 4.1. Dataset

The images collected to test the filter's performance are from the nuImages dataset. nuImages is a dataset of 93000 2d annotated images from a larger pool of data (nuScenes dataset). The images we used are randomly selected sample images from nuImages. We chose two images per camera as it allows us to evaluate the VQA model's ability to comprehend changes in perspective resulting from different camera angles. This approach ensures a diverse range of viewpoints, enabling a comprehensive assessment of the model's performance across various perspectives.

### 4.2. VQA Model: LXMERT

LXMERT (Learning Cross-Modality Encoder Representations from Transformers) is a large-scale Transformer model that consists of three encoders: an object relationship encoder, a language encoder, and a cross-modality encoder [11]. The model uses the Adam optimizer with a linear-decayed learning rate schedule and a peak learning rate at $1e - 4$. The model is trained for 20 epochs which is roughly 670K4 optimization steps with a batch size of 256. The pretraining of VQA tasks, however, is only for the last 10 epochs because this task converges faster and empirically needs a smaller learning rate [11]. An illustration of the networks in LXMERT is shown in Figure 4.

The VQA architecture in LXMERT facilitates comprehensive question-answering by integrating language and visual inputs. Using transformer layers of self-attention and cross-attention respectively, the model encodes contextual information from both textual queries and holistic visual features extracted from images. Through the collaborative operation of these components, with Lxmert Visual Feature Encoder and Lxmert Encoder, the model achieves a holistic understanding of the interplay between language and visual information to generate answers. However, the holistic approach of visual features is not necessarily a great idea when we want the model to only answer driving-related queries (examples in Supplementary Material).

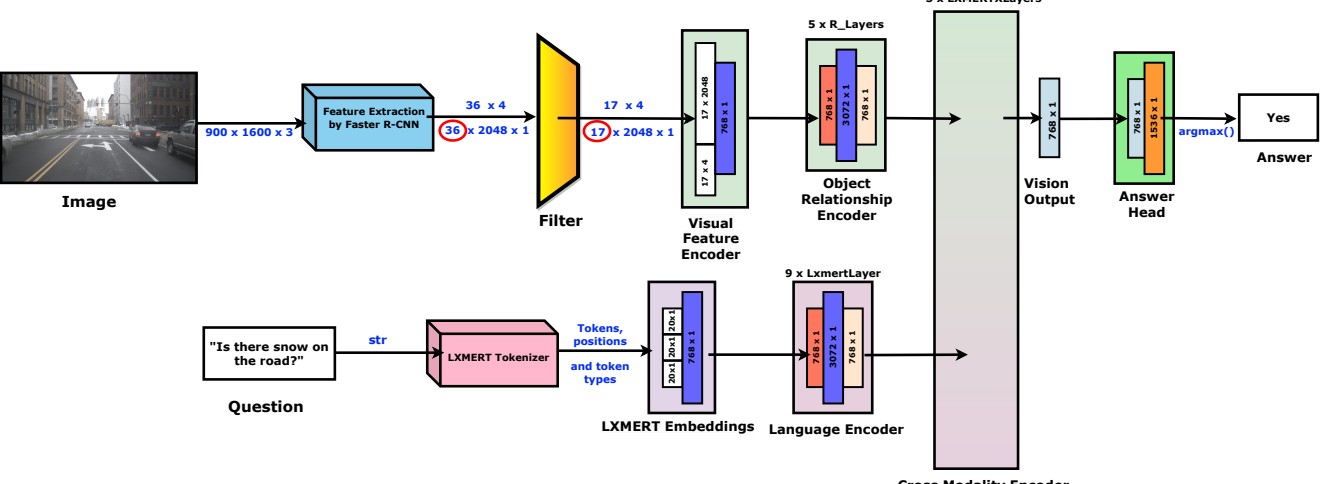

Figure 3. Visualizing the Functionality of a LXMERT with the filter integrated: An Illustrative Approach

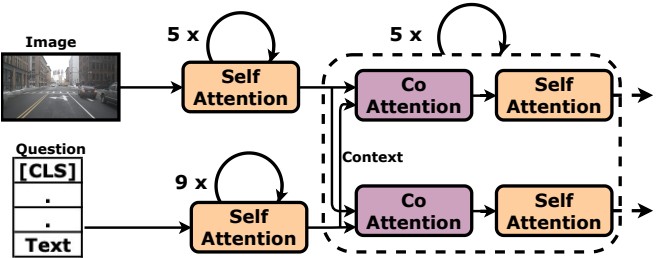

Figure 4. An Illustration of the architecture of LXMERT: self-attention with co-attention encoder

When we look deeper into the architecture of LXMERT, the input dimensions start with the image input, sized at 900 x 1600 pixels with 3 channels (RGB). The feature extraction of the image inputted to LXMERT is done using Faster R-CNN [2], where features are taken in 36 x 2048 x 1 and the boxes are taken in 36 x 4 dimensions. The filter we propose takes outputs from Faster R-CNN used in LXMERT along with parameters like device and detection threshold. It processes these outputs to extract relevant information such as predicted classes, scores, bounding boxes, normalized bounding boxes, and region of interest (ROI) features. It filters out predictions based on a predefined set of classes (e.g., signs, curbs, people, vehicles, etc.) using a detection threshold (circled in Red), which is 17 x 2048 x 1 dimensions. The function then returns the filtered information including filtered bounding boxes, classes, labels, indices, normalized bounding boxes, and ROI features which becomes the input for the Lxmert Visual Feature Encoder as shown in the Figure 3. These dimensions undergo transformations through convolutional layers and pooling layers resulting in higher-level feature representations (eg: 17 x 2048 and 3072 x 1) while reducing spatial dimensions to 768 x 1 as shown in Visual Feature Encoder and Object Relationship Encoder. This approach allows for the model to learn complex and abstract representations in the intermediate layers with 3072 features, potentially capturing more nuanced information or patterns. Then, by reducing the dimensionality back to 768 in the subsequent layers (R_Layers), the model can consolidate and distil this information into a more compact representation suitable for further processing or downstream tasks. Therefore, even though the input to the Cross modality Encoder has fewer features (768), the attention mechanism can still effectively capture relationships and dependencies across the input sequence.

Meanwhile, the question input is initially represented as word embeddings by taking tokens, positions, and token types as the input. It undergoes text processing in the Language Encoder block to capture the semantic information of the question. This process transforms the input question into a fixed-length vector representation. After separate processing of the image and question inputs, their features are combined in the Cross Modality Encoder enabling the model to leverage both visual and textual information. This joint representation retains relevant information from both modalities, facilitating the capture of complex patterns in the data. Subsequent layers, including the Vision Output layer (768 x 1), further process these combined features to capture intricate relationships between visual and textual cues. Finally, a probability distribution over possible answers, with dimensions corresponding to the number of answer classes in the dataset (1536) is processed in the Answer Head block. The final output answer with the most probability is chosen using argmax().

Table 1 is intended to show the difference in answers between an LXMERT model with and without the filter for better readability. The column features correspond to the features observed by the model when it generated that respective answer.

Table 1. Analyzing Feature Detection: LXMERT Pretrained Model with and without Filter vs. Human Observations

| Camera | Image | Questions | Human Answers | Features | Pretrained Answers | Features | Filter added | Features |
|---|---|---|---|---|---|---|---|---|
| Back Camera | | How many vehicles are there? | 3 | cars or bikes | 1 | tree, building, clouds, truck, road, pole, crosswalk, lines, sidewalk, scene | 0 | truck, road, crosswalk, lines, sidewalk |
| | | Which camera is this image from? | Back Camera | car behind | front | | unknown | |
| | | Are there any vehicles in ego lane? | No | vehicles in the lane | Yes | scene, sky, street, light, headlights, road | No | road, car, line |
| | | Is it safe to initiate a lane change? | unable to tell | not enough information | Yes | | No | |
| Back Left Camera | | Which camera is this image from? | unable to tell | Road Signs | front | sky, tree, pole, building, road, shadow, line, sidewalk, person | top | road, pole, line, person |
| | | Is it okay to initiate a lane change? | No | continuous white line | Yes | | No | |
| | | Are there any vehicles coming behind? | unable to tell | not enough information | Yes | tree, sky, pole, leaves, sign, building, grass, road, vehicle, line, sidewalk, bottle | No | road, pole, sign , line |
| | | Which street is this? | Summer Street | Name plate | unknown | | unknown | |
| Back Right Camera | | Which camera is this image from? | Back Right | road edges and markings | unknown | sky,tree, building, sign ,street, sidewalk, crosswalk, vent, tire, street, car, road, pole | unknown | road, truck, crosswalk, sign, pole, car |
| | | Do I need to stop? | No | already halfway in the turn | No | | No | |
| | | Are there any pedestrians on the sidewalk ? | No | sidewalk and pedestrians | No | stret, sidewalk, building, tree, road, bike, bus, door | No | road |
| | | Can I park on the right? | No | No parking slots | No | | No | |
| Front Camera | | Which camera is this image from? | Front Camera | road markings and vehicles in the front | front | building, water, road, car, lines, street, city | top | road, lines, water, car |
| | | Is there snow on the road? | Yes | Road and kerb | No | | No | |
| | | Are there any pedestrians? | Yes | pedestrians | No | building, tree, street, road, sidewalk, line, van | No | road, lines, line |
| | | Can I go right in this lane? | Yes | junction road to the right | No | | No | |
| Front Left Camera | | Which camera is this image from? | Front Left | kerb and pedestrians | front | ceiling, tree, building, pole, people, ground, sidewalk, line, man, window, person | unknown | pole, person, man, line |
| | | Can I take a left from here or should I go straight? | Don't know | not enough information | Yes | | Yes | |
| | | How many pedestrians are there? | Ten | Pedestrians | 0 | sky, tree, pole, building, sign, woman, crosswalk, road, median, line, shirt, pants, man, people, person | 0 | road, crosswalk, person, sign, line, man |
| | | Do I need to stop till pedestrians cross to turn left? | Yes | vehicle orientation and pedestrians | No | | No | |
| Front Right Camera | | Can I park here? | No | parking slots | No | building, bus, van, circle, car, sign, tire, ceiling | No | car, sign |
| | | Why can't I park here? | No space | empty slots not available | parking | | No | |
| | | Which camera is this image from? | Front Right | sign boards and directions | front | grass, road, curb, man, tree, sign, sky, building | unknown | road, sign, man, curb, pole, person |
| | | Which direction can I drive in? | Only straight | road markings and kerb | right | | right | |

It can be seen from the 'Camera' column that we tried to keep diverse driving scenarios in mind while designing the case study. The answers received from LXMERT, both pretrained and when the filter has been integrated, have been listed along with the features extracted in each case (to the right of the corresponding column). It provides a visual representation of the model's performance in addressing the posed questions, allowing for an assessment of their effectiveness based on the Human Answers. The reason for comparing the outputs of three VQA models with human answers, using colour coding (green for correct, red for wrong, yellow for partially correct), is to visually emphasize performance and discrepancies between the models and human responses. This visual representation allows for a quick and intuitive understanding of the accuracy and effectiveness of the models in comparison to human performance. Further discussion of this rationale is considered in the paper [10] and the results in the table are discussed in 5.

## 5. Results and Discussion

We use the subjective scoring framework for VQA models [1] in autonomous driving to gauge the improvement of LXMERT after the filter has been added. This scoring system analyses the answers provided by the VQA model using multiple types of natural language processing models (BERT-base-uncased, NLI-distilBERT-base, all-mpnet-base-v2 and GPT-2) [4] and sentence similarity benchmark metrics (Cosine Similarity) [6]. The results are shown in the Table 2.

It can be observed from Figure 5 that there is a noticeable enhancement in the model's performance after the integration of the filter as the MAE (Figure 5a) and RMSE

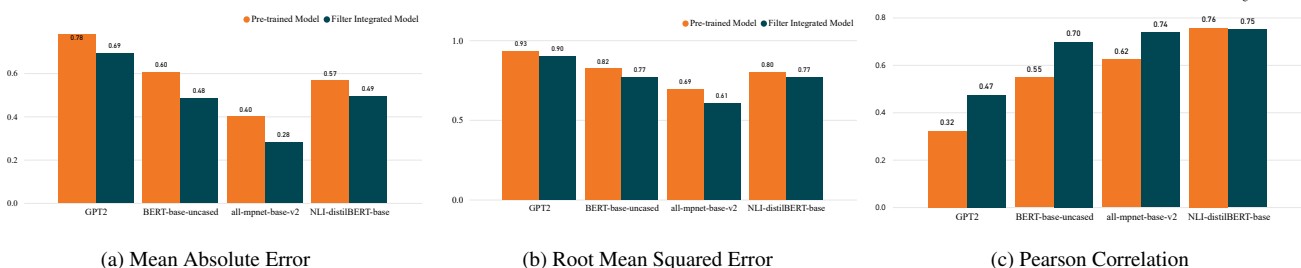

(a) Mean Absolute Error        (b) Root Mean Squared Error        (c) Pearson Correlation

Figure 5. Assessment of LXMERT using the Subjective Scoring Framework

Table 2. Evaluation of LXMERT Performance using Subjective Scoring Framework Metrics: MAE, RMSE, and Pearson Correlation

| LXMERT model | Mean Absolute Error | | Root Mean Squared Error | | Pearson Correlation | |
|---|---|---|---|---|---|---|
| | Pretrained | Filtered | Pretrained | Filtered | Pretrained | Filtered |
| NLI-distilBERT-base | 0.5660 | 0.4942 | 0.7998 | 0.7712 | 0.7558 | 0.7504 |
| all-mpnet-base-v2 | 0.3989 | 0.2802 | 0.6932 | 0.6068 | 0.6231 | 0.7370 |
| BERT-base-uncased | 0.6042 | 0.4840 | 0.8229 | 0.7675 | 0.5480 | 0.6968 |
| GPT2 | 0.7778 | 0.6931 | 0.9344 | 0.9010 | 0.3220 | 0.4737 |

(Figure 5b) scores have lowered when compared to the pre-trained model. The increase in Pearson correlation (Figure 5c) scores shows that the answers given by the filter-integrated model are closer to the human answers which is ultimately the goal for any VQA model. However, it has to be acknowledged that there are erroneous responses despite this enhancement. These inaccuracies are due to the inherent limitation of the VQA model, as it was not originally designed or trained specifically for driving-related queries.

To address this discrepancy, fine-tuning the model with a driving dataset is a viable solution. This process of fine-tuning will equip the model with the necessary contextual knowledge to interpret questions from a driving perspective accurately, consequently refining its responses accordingly.

After integrating the filter, it's evident from the observed features (Figure 1) that the model has begun to emulate human attention patterns to a remarkable extent. This enhancement is significant for its ability to focus on relevant information. By aligning more closely with human attention patterns, the model becomes more adept at understanding nuanced context, discerning subtle cues, and prioritizing relevant data points. This heightened cognitive alignment improves the model's interpretability and enhances its adaptability in driving scenarios. We further show examples of a few cases in the Supplementary Material where we observe in the figures the differences in object detection and the model's answers due to different filter weights at the Feature extraction stage.

## 6. Conclusion and Future Work

In conclusion, this study has introduced a novel filter designed to enhance the performance of VQA models specifically in driving-related tasks. Through our case study, we have demonstrated the efficiency of the filter in mimicking human attention patterns to a significant extent, thereby laying the groundwork for improved VQA capabilities. The limitation of this approach is that we assume that the human is telling what they are actually observing which is leaving a scope for subjectivity in data. For future experiments, we would like to use the eye tribe tracker that delivers real-time data of where a person is looking on a screen similar to [8]. This would potentially improve the accuracy and reliability of the observations in future experiments. However, it's essential to acknowledge that VQA models are not inherently trained for driving tasks, highlighting the need for further optimization and adaptation. Our future work will focus on fine-tuning at least three VQA models using an exclusive driving dataset such as Nuscenes MQA [3], tailored to the complexities of driving environments. By training VQA models on annotated driving scenes and questions, we aim to bolster their performance and adaptability in addressing driving-related queries. Additionally, we plan to conduct a thorough analysis of the fine-tuned models' performances to gain insights into the effectiveness of model adaptation. We also intend to explore the integration of a layer capable of understanding camera information into VQA models. This enhancement will enable the models to perform spatial reasoning tasks more effectively, analyze object positioning within the camera frame, and provide dynamic and adaptive responses to queries about the driving environment. By configuring the filter so that it is capable of leveraging camera information, we aim to bridge the gap between human and machine attention patterns, thereby advancing the capabilities of VQA models in driving scenarios.

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
