# OpenReview forum: "Optimizing Visual Question Answering Models for Driving: Bridging the Gap Between Human and Machine Attention Patterns"
_thecvf.com/CVPR/2024/Workshop/VLADR — VLADR 2024 Poster_

### Official Review · Reviewer_Y9Xt · 2024-04-17
**Interesting problem but there's much room for improvement.**

**Rating:** 6
**Confidence:** 4

**Review:**

Summary:

The paper investigates how VQA models can be optimized for autonomous driving applications. The study identifies disparities in the attention patterns between humans and a VQA model, specifically how each focuses on different objects when answering driving-related questions. The main contribution of the paper is the development of a filter that integrates into the VQA model to prioritize relevant objects and improve the accuracy of responses. This is achieved by comparing the attention patterns of humans and both pre-trained and filter-integrated models using the LXMERT model and images from the NuImages dataset. The results show that the feature encoder filter enhances the model's performance by refining its attention mechanisms.

Strengths:

The difference in attention mechanism of VQA models and real humans is a fundamental problem to reach better alignment in different downstream tasks. This paper provide good insight on such differences in the context of autonomous driving.

The architecture of the methodology of the proposed LXMERT method is clear, allowing better reproducibility.

Weaknesses:

"we surveyed ten individuals with a minimum of five years of driving experience" This number should be increased to avoid any bias.

The proposed filter in Algorithm 1 does not seem to be generalizable to long-tail cases, since it is simply a hard filter for certain classes.

Tables 1 shows there are many failure cases even after adding the filter.

The framework show in Figure 3 does not look applicable to most recent VLMs like GPT-4V or Gemini.

---

### Decision · Program_Chairs · 2024-04-22

Accept (Poster)